# Behavioral and electrophysiological responses of *Hippodamia variegata* to plant volatiles

Yi Wang[1,2], Ying Zhang[2], Tao Zhang[3], Zihan Li[1,2], Tinghui Liu[1]*, Yanhui Lu[2]*

1 College of Plant Protection, Hebei Agricultural University, Baoding, China, 2 State Key Laboratory for Biology of Plant Diseases and Insect Pests, Institute of Plant Protection, Chinese Academy of Agricultural Sciences, Beijing, China, 3 State Key Laboratory of IPM on Crops in Northern Region of North China, Institute of Plant Protection, Hebei Academy of Agriculture and Forestry Sciences, Baoding, China

* liutinghui@126.com (TL); luyanhui@caas.cn (YL)

## Abstract

Natural enemies use odors to find prey in nature. The use of plant volatiles to attract natural enemy insects and promote their feeding on pests has great potential for practical application in integrated pest management. Of the 28 materials tested, ethyl benzoate, octyl formate, methyl jasmonate, and methyl dihydrojasmonate were significantly attractive to *H. variegata*. We tested the ladybeetle's behavioral responses to three concentrations of the compounds (100, 10, and 1 μg/μL) and found that high and medium levels were both attractive to *H. variegata*. In electroantennography (EAG) tests, the response of *H. variegata* to the above four compounds increased with the increasing concentration. Also, feeding of *H. variegata* increased in the presence of these plant volatiles under the condition of enclosures. These findings provide a theoretical basis for future behavioral manipulation of this predator under field conditions.

## 1. Introduction

In complex ecosystems, only when natural enemies find pests in an accurate and timely manner will the natural enemy establish and increase in density [1]. Predators find prey in a variety of ways. For example, prey may be detected by the predator through its senses of smell [2], vision [3], or tactile feelings [4]. One of the most important of the insect's senses is the use of smell to perceive odors linked to prey. Natural enemy insects use smell to find prey based on odors released by the pest itself [5], odors produced by the host plant [6], or pest-induced plant volatiles (HIPVs) [7]. Plants release a series of chemical messages when stimulated by the external environment or when they bloom and bear fruit [8], The chemical information in these substances can participate in the chemical information exchange between natural enemies and plants [9]. When such odors are detected, the experience can stimulate behavioral responses in the pest [10], or they may attract natural enemies of the pest [11], or they may cause the pest to avoid the natural enemy [12]. For example,

**Data availability statement:** All relevant data are within the paper and its Supporting Information files.

**Funding:** This work was supported by the National Key R&D Program of China (2022YFD1400300), and China Agriculture Research System of MOF and MARA (CARS-15-19).

**Competing interests:** The authors have declared that no competing interests exist.

aphid-infested marigolds can attract *Harmonia axyridis* (Pallas) [13]. Similarly, when offered an aphid-infested plant versus an uninfested plant, adults of the ladybeetle *Coccinella septempunctata* Linnaeus are significantly more attracted to the infested plant's odor, and the beetles show greater prey finding and feeding activity [14].

Volatile organic compounds (VOCs) come in many varieties [15]. However, the main types that strongly attract natural enemies are alcohols, esters, terpenes, aromatic groups, and ketones [16]. VOCs are volatile secondary substances continuously released by plants during their growth and development [17], and play an important role in plant information transmission [18]. At present, an increasing number of volatile compounds are being used to attract natural enemy insects for biological control [19]. For example, high concentrations of geraniol (10 and 0.1 g/L) have a higher attraction rate for adult *Orius nagaii* Yasunaga [20]. Similarly, the behavior of insect natural enemies can be effectively manipulated by mixing relevant compounds. For example, a mixture of methyl salicylate, farnesene, and benzaldehyde can be more effective at attracting *Hippodamia variegata* [21]. Also, adult *H. axyridis* are attracted to a mixture of 1, 2-diethylbenzene and p-diethylbenzene [22]. Several mixed formulations of (E) -β-farnesene, methyl anthranilate and (-) -β-caryophyllene have significant attractant effects on adult *H. axyridis* [23]. A ternary mixture of phenylacetonitrile, 2-phenylethanol, and acetic acid attracted many more *Chrysoperla plorabunda* (Fitch) than a single component [24]. In general, using VOC mixtures to protect boundary vegetation allows for greater and more accurately located biological control of pests along crops boundaries [25].

*H. variegata* is an important natural enemy of aphids found on many crops [26], which mainly feeds on *Aphis gossypii* Glover [27], but which also preys on the *Poratrioza sinica* Yang et Li [28], the aphid *Acyrthosiphon pisum* (Harris) [29], and eggs and young larvae of some lepidoptera pests. *H. variegata* has many advantages for pest control, such as a wide range, common occurrence, high predation raters [30], and high resistance to environmental stress [31]. It is an important predator in northwest China [32]. Previous studies have shown that volatile compounds such as methyl salicylate, farnesene, and benzaldehyde can attract *H. variegata* [7,21,33]. These findings indicate that plant-derived semiochemicals have promising potential for the behavioral manipulation of *H. variegata*.

Based on previous research and the application potential of plant-derived volatile compounds in insect behavioral manipulation, we selected 28 representative volatile compounds in this study to evaluate their effects on the behavior of *H. variegata*. These compounds included those previously reported to attract other coccinellid species, such as methyl jasmonate [34]. We also included their structural analogs, such as methyl dihydrojasmonate. In addition, we selected compounds known to repel aphids, such as volatiles from mint (*Mentha* spp.), including menthol and menthone [35]. We also included commonly occurring floral and fruity volatiles, such as octyl formate, nonanol, and phenethyl alcohol, which may trigger olfactory responses due to the flower-visiting behavior of *H. variegata* [36,37]. All selected compounds are plant-derived and widely distributed in plant–aphid–predator interaction systems. They are closely associated with the natural foraging habitat of *H. variegata*,

providing strong biological relevance. This study aims to systematically assess the behavioral responses of *H. variegata* to 28 plant volatiles with potential attractant activity. First, we used a Y-tube olfactometer to screen for the effective compounds, and we then examined the efficacy of different concentrations of the most attractive materials to identify the optimal concentration. Second, we used electroantennography (EAG) tests to observe the antennal potential response to the compounds with the best performance in the Y-tube tests. Finally, we ran a cage test to determine if the best-performing compounds would promote the feeding effect of *H. variegata* in a caged arena.

## 2. Materials and methods

### 2.1 Source of insects for tests

The insects we used in this study were from a laboratory colony of *H. variegata* that was established with insects collected in 2022 from experimental cotton fields (CAAS, 39.53°N, 116.70°E) of the Langfang Experiment Station of the Chinese Academy of Agricultural Sciences (Hebei Province, China). *Hippodamia variegata* were fed on *Myzus persicae* (Sulzer) at (25±1°C, 65%±5% RH, and 16:8 L:D photoperiod. *Myzus persicae* was reared on pea seedlings held at 19±1°C, 65±5% RH, and 16:8 L:D photoperiod. Pea seedlings were grown in square plastic pots (15×15×15 cm) filled with vermiculite. Pea seedlings were planted at high density to provide a suitable environment for aphid growth and reproduction.

### 2.2 Sources of test chemicals

Table 1 provides details on the purity and sources of the 28 compounds and one solvent (liquid paraffin) used in our study.

### 2.3 Y-tube olfactory tests

Behavioral responses of *H. variegata* were evaluated using a Y-tube olfactometer, following the procedures of Yu et al [38], with slight modifications. Before the whole experiment began, filter paper impregnated with paraffin oil was placed on both arms of the Y-tube and assessed to ensure there was no inherent bias between the left and right sides.

Each compound was tested on 60 male and 60 female adults of *H. variegata*. Active, healthy adults of *H. variegata* (both sexes) were selected and starved for 24 h before tests. Each test was carried out at 25±1 °C between 9 a.m. and 5 p.m. [7] under conditions similar to colony maintenance. A QC-3 atmospheric sampler was used as the air-flow power source, and a filter with activated carbon, a gas-washing bottle with ultrapure water, a glass rotameter, an odorant bottle, and the two arms of a Y-tube (inner dia 1.5 cm, base 18 cm long, and arms (9 cm long with a 60° angle) were connected in sequence. The parts of the olfactometer were connected to each other using Viton tubing, and the connections were sealed with sealing film to prevent air leakage. Each volatile compound was diluted to 10 µg/µL with liquid paraffin, and 10 µL were pipet-ted onto a piece of filter paper (1.5×1.5 cm). The test used liquid paraffin as the control. For each run, we place a treatment-impregnated filter paper in one odor bottle and a control filter paper in the other. To begin a test, we allowed treatment and control odors to move through the device, pulled by the gas flow whose flow rate was stably controlled at 400 mL/min. We allowed 5 min for the odors in the device to stabilize.

The lady beetles were released at the base of the central arm of the testing apparatus. When a beetle reached the halfway point of the main arm of the device, a stop-watch was pressed to begin timing. The behavior of each beetle was observed for a total of 5 min. If a beetle entered one of the two arms, moved up more than one-third the arm's length, and remained there for at least 5 s, it was recorded as having selected that odor source. If no selection was made within 5 min, the beetle was recorded as having made no selection. After every 5 insects tested, the positions of the left and right arms of the Y-tube were swapped to eliminate positional effects. After every 10 insects tested, the Y-tubes were replaced with new ones. The Y-tubes were cleaned once with alcohol and three times with distilled water before use and were placed in an oven at 120 °C for two hours. Each insect was tested only once, and after the test, the Y-tube, the flavor source bottle, and the Teflon tube at the connection were washed with alcohol and dried.

**Table 1. Standardized information on substances used in study.**

| No. | Standard compounds | CAS | Molecular formula | Attacking coefficient |
|---|---|---|---|---|
| 1 | Menthol | 89-78-1 | $C_{10}H_{20}O$ | Aladdin Biochemical Technology Co., Ltd (Shang-hai, China) |
| 2 | Menthone | 10458-14-7 | $C_{10}H_{18}O$ | Aladdin Biochemical Technology Co., Ltd (Shang-hai, China) |
| 3 | 1-Octene | 111-66-0 | $C_8H_{16}$ | TCI Development Co., Ltd (Shanghai, China) |
| 4 | Terpineol | 8000-41-7 | $C_{10}H_{18}O$ | Aladdin Biochemical Technology Co., Ltd (Shanghai, China) |
| 5 | Neomenthol | 2216-52-6 | $C_{10}H_{20}O$ | TCI Development Co., Ltd (Shanghai, China) |
| 6 | iso-Mentone | 491-07-6 | $C_{10}H_{18}O$ | TCI Development Co., Ltd (Shanghai, China) |
| 7 | Sclareol | 515-03-7 | $C_{20}H_{36}O_2$ | TCI Development Co., Ltd (Shanghai, China) |
| 8 | Myrcene | 123-35-3 | $C_{10}H_{16}$ | Aladdin Biochemical Technology Co., Ltd (Shanghai, China) |
| 9 | Phenylethyl alcohol | 60-12-8 | $C_8H_{10}O$ | Aladdin Biochemical Technology Co., Ltd (Shanghai, China) |
| 10 | Nerol | 106-25-2 | $C_{10}H_{18}O$ | Aladdin Biochemical Technology Co., Ltd (Shanghai, China) |
| 11 | α-Terpinene | 99-86-5 | $C_{10}H_{16}$ | InnoChem Science & Technology Co., Ltd. (Beijing, China) |
| 12 | β-Eudesmol | 473-15-4 | $C_{15}H_{26}O$ | Aladdin Biochemical Technology Co., Ltd (Shanghai, China) |
| 13 | Hexanal | 66-25-1 | $C_6H_{12}O$ | TCI Development Co., Ltd (Shanghai, China) |
| 14 | 1-Nonanol | 143-08-8 | $C_9H_{20}O$ | Aladdin Biochemical Technology Co., Ltd (Shanghai, China) |
| 15 | 1-Nonanal | 124-19-6 | $C_9H_{18}O$ | Aladdin Biochemical Technology Co., Ltd (Shanghai, China) |
| 16 | Ocimene | 13877-91-3 | $C_{10}H_{16}$ | TCI Development Co., Ltd (Shanghai, China) |
| 17 | Pentadecane | 629-62-9 | $C_{15}H_{32}$ | Aladdin Biochemical Technology Co., Ltd (Shanghai, China) |
| 18 | Methyl benzoate | 93-58-3 | $C_8H_8O_2$ | Aladdin Biochemical Technology Co., Ltd (Shanghai, China) |
| 19 | p-Anisaldehyde | 123-11-5 | $C_8H_8O_2$ | TCI Development Co., Ltd (Shanghai, China) |
| 20 | 1,2-Diethylbenzene | 135-01-3 | $C_{10}H_{14}$ | Aladdin Biochemical Technology Co., Ltd (Shanghai, China) |
| 21 | Methyl heptenone | 110-93-0 | $C_8H_{14}O$ | Aladdin Biochemical Technology Co., Ltd (Shanghai, China) |
| 22 | 3-Ethylacetophenone | 22699-70-3 | $C_{10}H_{12}O$ | Aladdin Biochemical Technology Co., Ltd (Shanghai, China) |
| 23 | Butyl acrylate | 141-32-2 | $C_7H_{12}O_2$ | Aladdin Biochemical Technology Co., Ltd (Shanghai, China) |
| 24 | Methyl dihydrojasmonate | 24851-98-7 | $C_{13}H_{22}O_3$ | TCI Development Co., Ltd (Shanghai, China) |
| 25 | Octyl formate | 112-32-3 | $C_9H_{18}O_2$ | Aladdin Biochemical Technology Co., Ltd (Shanghai, China) |
| 26 | Ethyl benzoate | 93-89-0 | $C_9H_{10}O_2$ | TCI Development Co., Ltd (Shanghai, China) |
| 27 | Methyl jasmonate | 39924-52-2 | $C_{13}H_{20}O_3$ | Yuanye Bio-Technology Co., Ltd (Shanghai, China) |
| 28 | Thymol | 89-83-8 | $C_{10}H_{14}O$ | Aladdin Biochemical Technology Co., Ltd (Shanghai, China) |
| 29 | Liquid paraffin | 8042-47-5 | / | Sigma-Aldrich, St Louis, MO, USA |

## 2.4 Effects of different concentrations of tested compounds

The plant volatiles that were attractive to *H. variegata* in the initial Y-tube tests were formulated as three different concentrations: 100, 10, and 1 µg/µL for further behavioral assays [53]. Each compound was tested at three different concentrations on 60 male and 60 female adults of *H. variegata*. The assay procedure was the same as described above; after the assay was completed, the results were compared, and the concentration giving the greatest response was identified for further EAG assays.

## 2.5 Electroantennography (EAG) testing of best compounds

The plant volatiles (and concentrations) eliciting the strongest response in the Y-tube assays were further tested measuring electroantennography (EAG) activity. EAG tests were run using antennae of 3–5 days old female and male adults of *H. variegata*.

A glass capillary (0.2 mm inner dia) was used as the electrode, after it was pulled out with a P-1000 needle puller to produce a tip of suitable length and caliber. Two such capillary tips were then cut with a blade to make a suitable-sized (dia), and then a saline solution was drawn up into the tip to the 1/2 mark of the point for later use.

The antennae of an *H. variegata* adult were cut off from the head and attached at the base of the antenna to the cut end of the reference electrode. Then the tip of the antennae was attached to the cut end of the recording electrode. The two glass electrodes, once they were joined as just stated to the antenna of the test beetle, were then inserted into silver-silver chloride electrodes fixed to the micromanipulator.

A strip of filter paper (0.4 cm × 3.0 cm) impregnated with 10 μL of the test compound was inserted into a Pasteur pipette, which was then connected to a precision gas flow control apparatus. The continuous gas flow rate was set at 124 mL/min, and the stimulation gas flow rate was set at 20 mL/min. The stimulation odor source was located about 1 cm from the antennae, and the time of stimulation for each test was 0.2 s. The interval between two stimulations was 30 s. Each compound was tested with the above method. Six antennae were tested for each compound, and each antenna was exposed to the solvent (mineral oil) before and after testing specific compounds.

At the end testing the 28 compounds of interest, we made direct comparisons of several compounds to see if they elicited greater or less responses by antennae.

### 2.6 The impact of effective volatiles on the predation efficiency of *H. variegata*

The plant volatiles that have been assayed (Y-tube or EAG) for their attractive effects on *H. variegata* were next tested in a caged, potted plant assay. This test was run daily at 9 a.m. in an insect cage (a cube with 1 m sides) covered with 80-mesh nylon gauze. In the test cage, we placed two, 4-leaf, potted cotton plants diagonally across from each other inside the cage (The pot is 15 cm in diameter), and 100 wingless *A. gossypii* were placed on each of the two cotton plants before the test. A wooden chopstick was then infixed into each of the two pots, and a 1 × 2.5 cm piece of filter paper was glued on the top of each chopstick. Subsequently, 20 μL of the test compound (10 μg/μL) was dropwise applied onto filter paper positioned 10 cm above the soil surface of an individual plant, while 20 μL of liquid paraffin was added to the filter paper on the other plant as the control. The insect cage was then placed in a greenhouse with the same light, humidity, temperature, and one robust *H. variegata* was placed on a Petri dish in the center of the cage. We then counted the number of aphids remaining on each cotton plant 24 h later. Six cages were set up in this manner for each compound to be tested, with each cage being treated as a replicate.

### 2.7 Data analysis

Microsoft Excel (Microsoft, Redmond, WA, USA) and SPSS version 26.0 software (IBM Corporation, Armonk, NY, USA) were used for statistical analysis. Graphs were created using GraphPad Prism 8.0 (GraphPad Software, La Jolla, CA, USA). Data from the olfactory behavioral selection test for different compounds' odors and the results of behavioral tests with different compounds were analyzed using Chi-square tests. $\chi^2$ and p-values were calculated, after excluding non-responsive beetles. For EAG test data, we used one-way analysis of variance (ANOVA) with Duncan multiple range test, and we used independent sample t-tests for data from the cage test of *H. variegata*'s response to target compounds. Where $p < 0.05$ was significant (*) and $p < 0.01$ (**) was highly significant. Response values from EAG tests were calculated as response value = response value – blank value (where the control value is the EAG response value of liquid paraffin).

## 3. Results

### 3.1. Attractive effects of test compounds for *H. variegata*

Among the 28 compounds tested, four were significantly attractive to adult males of *H. variegata*: ethyl benzoate ($\chi^2 = 4.571$, $p = 0.033$), octyl formate ($\chi^2 = 4.898$, $p = 0.027$), methyl jasmonate ($\chi^2 = 6.333$, $p = 0.012$), and methyl dihydro-jasmonate ($\chi^2 = 4.414$, $p = 0.036$). For adult females of *H. variegata*, the same four compounds were also attractive: ethyl benzoate ($\chi^2 = 3.947$, $p = 0.047$), octyl formate ($\chi^2 = 4.414$, $p = 0.036$), methyl jasmonate ($\chi^2 = 4.414$, $p = 0.036$), and methyl

dihydrojasmonate ($\chi^2=6.000$, $p=0.014$), while the rest of the compounds showed no significant attraction to either sex of *H. variegata* (Fig 1).

### 3.2. Attraction of different concentrations of test compounds to *H. variegata*

The four compounds that were attractive to *H. variegata* – methyl jasmonate, methyl dihydrojasmonate, ethyl benzoate, and octyl formate – were tested at different concentrations (Fig 2). At 100 µg/µL, all four compounds were attractive to *H. variegata*, with methyl jasmonate ($\chi^2=4.412$, $p=0.036$), methyl dihydrojasmonate ($\chi^2=4.245$, $p=0.039$), ethyl benzoate ($\chi^2=6$, $p=0.014$), and octyl formate ($\chi^2=5.255$, $p=0.022$) being significantly attractive to adult males, and ethyl benzoate ($\chi^2=16.667$, $p<0.001$), octyl formate ($\chi^2=12$, $p<0.001$) being highly significantly attractive to females, while methyl jasmonate ($\chi^2=4.083$, $p=0.043$) and methyl dihydrojasmonate ($\chi^2=6$, $p=0.014$) had a significant but less attraction to females.

At 10 µg/µL, *H. variegata* was attracted by the four compounds, with methyl jasmonate ($\chi^2=6.333$, $p=0.012$), methyl dihydrojasmonate ($\chi^2=4.414$, $p=0.036$), ethyl benzoate ($\chi^2=4.571$, $p=0.033$), and octyl formate ($\chi^2=4.898$, $p=0.027$) being significantly attractive to adult males, and ethyl benzoate ($\chi^2=3.947$, $p=0.047$), octyl formate ($\chi^2=4.414$, $p=0.036$), methyl jasmonate ($\chi^2=4.414$, $p=0.036$), and methyl dihydrojasmonate ($\chi^2=4.414$, $p=0.036$) being significantly attractive to adult females.

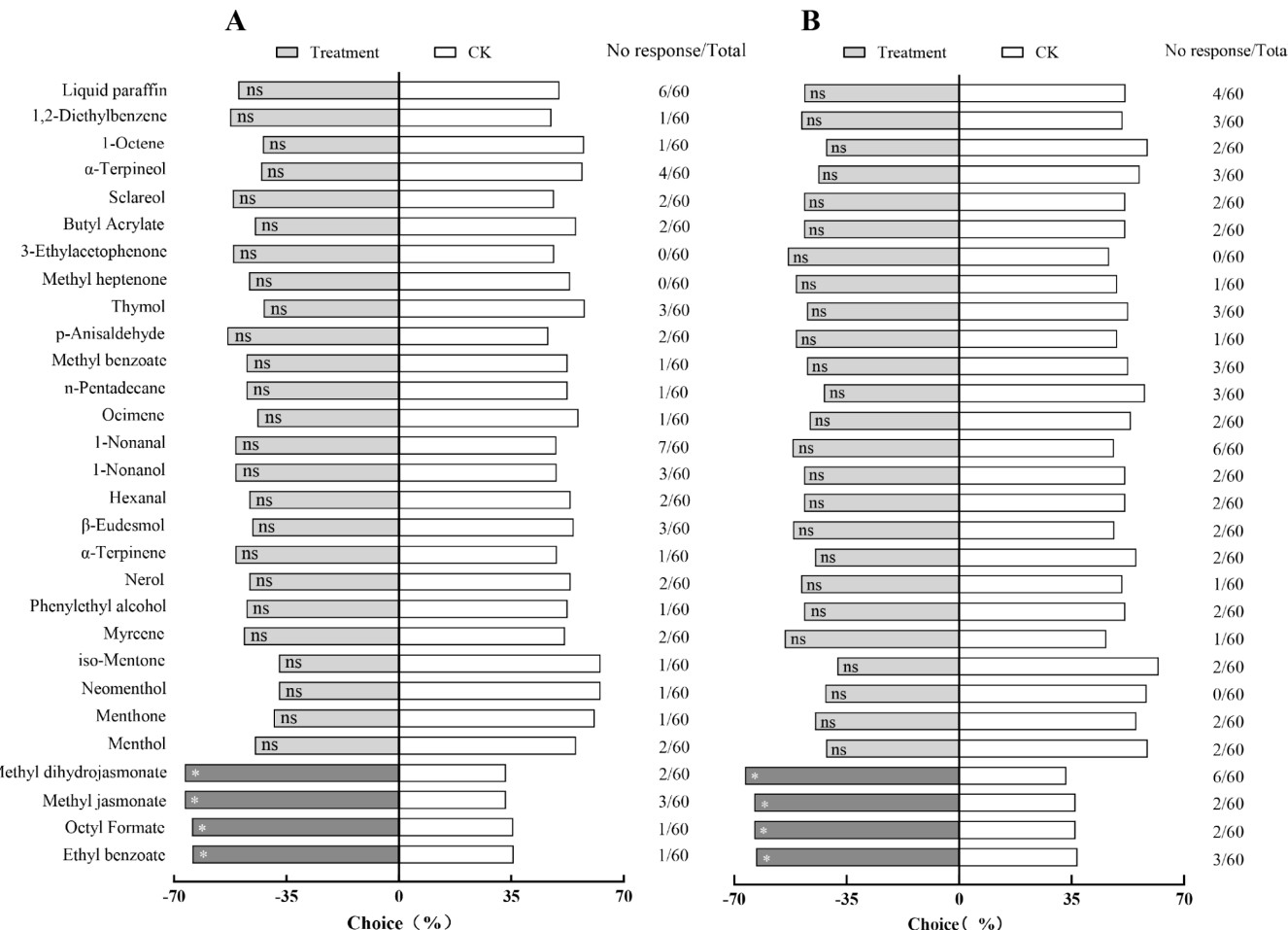

**Fig 1. Degree of preference of *H. variegata* for 28 compounds tested in a Y-tube olfactometer at a concentration of 10 µg/µL for males. (A) and females (B),** where asterisks (*) mark significant differences of $p<0.05$; ns indicates no significant differences.

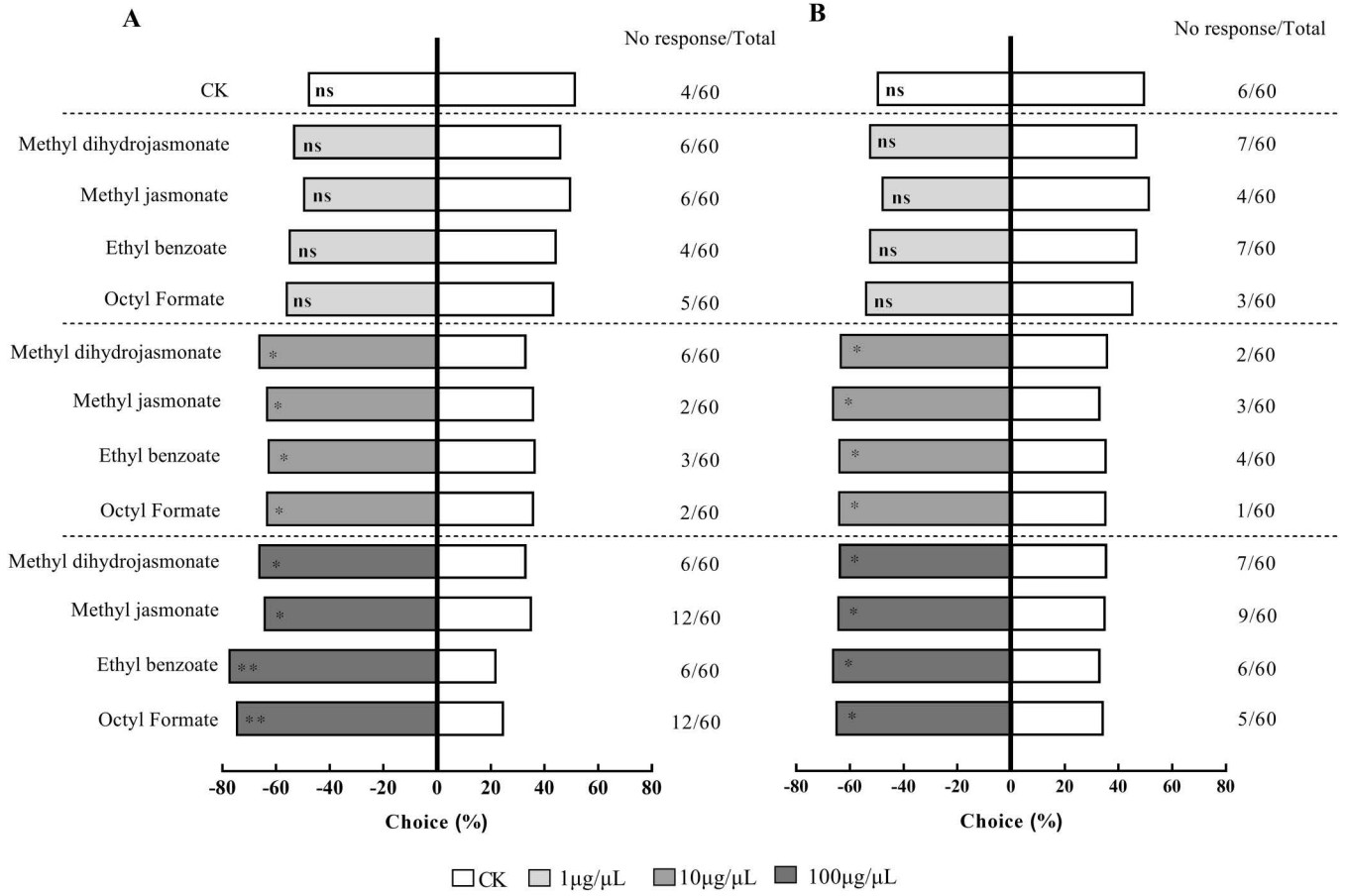

**Fig 2. Attraction of three concentrations of four compounds to adult females (A) and males (B) of *H. variegata*.** CK is the control of liquid paraffin; Asterisks (*) denote statistically significant differences at $p < 0.05$ and (**) denoting highly significant differences at $p < 0.01$; ns represents no significant differences.

At 1 µg/µL, none of the four compounds had any significant attraction for either male or female adults of *H. variegata.*

As shown in Fig 2, the effects of octyl formate and ethyl benzoate at 100 µg/µL are highly significant, with their attraction effects stronger than those at 10 µg/µL. By comparing their P-values, we found that methyl jasmonate and methyl dihydrojasmonate have better attraction effects on the male *H. variegata* at 10 µg/µL than at 100 µg/µL.

### 3.3. Electroantennography (EAG) responses of *H. variegata* to test compounds

Significant EAG responses were observed to octyl formate, ethyl benzoate, as well as smaller but still significant responses to methyl jasmonate and methyl dihydrojasmonate (Fig 3).

EAG responses of ethyl benzoate and octyl formate were significant at 10 µg/µL and 100 µg/µL for both females and males, while the EAG responses for methyl jasmonate and methyl dihydrojasmonate were significant only at 100 µg/µL, for both females and males.

### 3.4. Evaluation of the functional effects of predation by *H. variegata* in response to target compounds

In this experiment, methyl jasmonate ($\alpha = 0.615$, df $= 10$, $p = 0.002$) and octyl for-mate ($\alpha = 0.830$, df $= 10$, $p = 0.008$) had a highly significant enhancing effect on feeding by adult males of *H. variegata*, while a lesser but still significant effect was

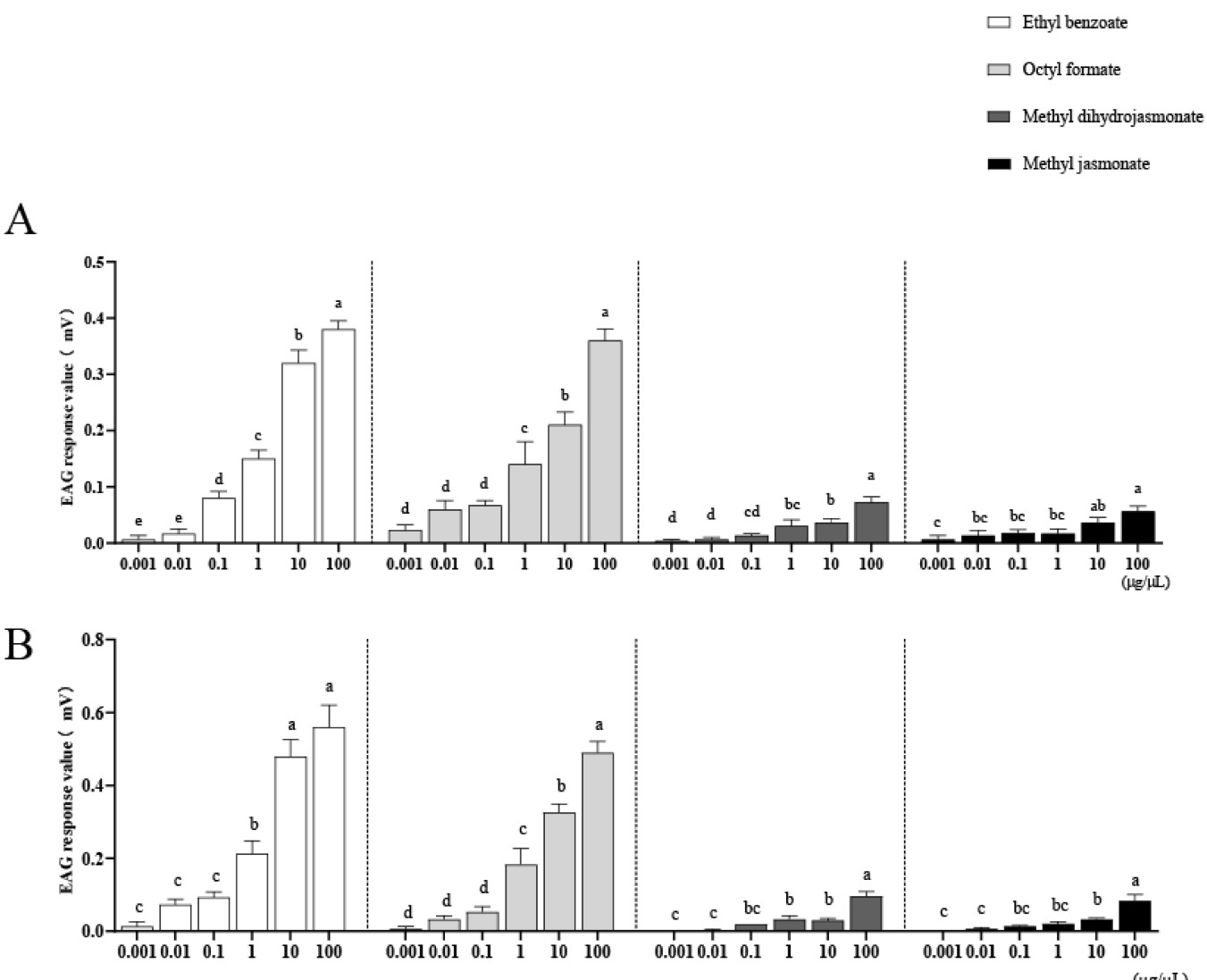

**Fig 3. EAG responses of adult males (A) and females (B) of *H. variegata* to four substances.** Different lowercase letters represent significant differences. Results are means ± SEM.

found in males for ethyl benzoate (α = 0.205, df = 10, *p* = 0.017) and methyl dihydrojasmonate (α = 0.593, df = 10, *p* = 0.013). For females, three compounds – methyl jasmonate (α = 0.091, df = 10, *p* = 0.002), ethyl benzoate (α = 0.926, df = 10, *p* < 0.001), and methyl dihydrojasmonate (α = 0.172, df = 10, *p* = 0.004) – had a highly significant enhancing effect on feeding, while octyl formate (α = 0.170, df = 10, *p* = 0.019) had a lesser, but still significant enhancing effect feeding (Fig 4).

## 4. Discussion

In this study, we found that ethyl benzoate, octyl formate, methyl jasmonate, and methyl dihydrojasmonate were attractive to *H. variegata*. When three concentrations (1, 10, 100 µg/µL) of the above compounds were examined, we found both the concentrations of 10 and 100 had a significant attraction effect on *H. variegata*. For octyl formate and ethyl benzoate, the concentration of 100 µg/L exhibited a stronger attractiveness to females compared to 10 µg/L, while no significant

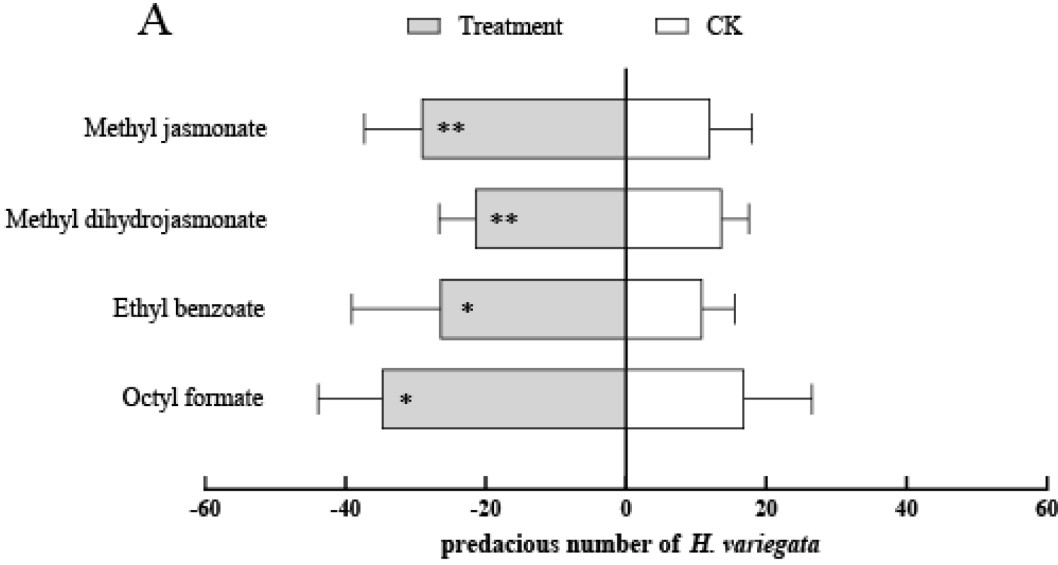

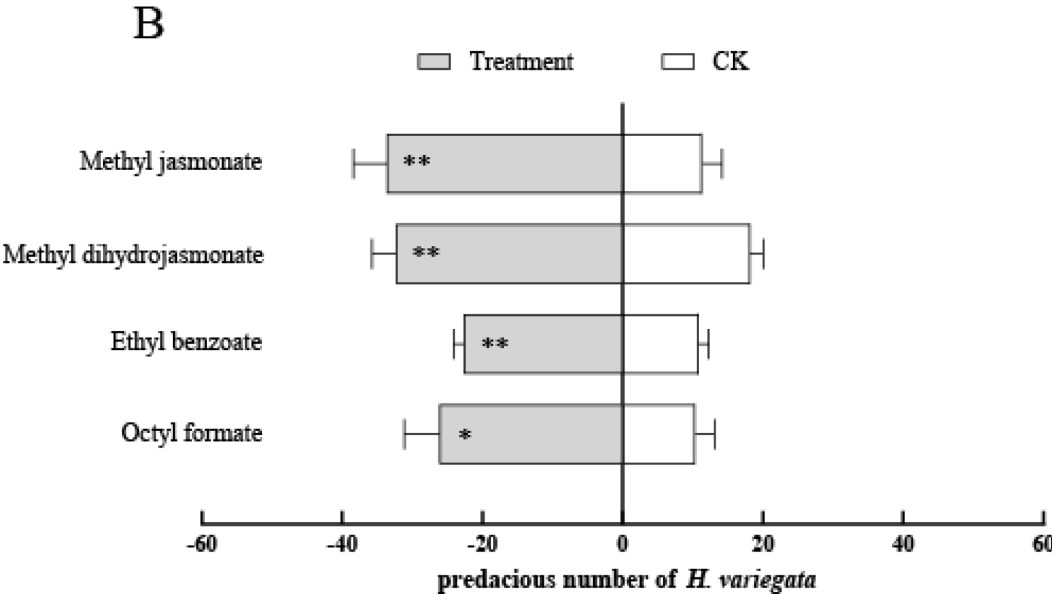

**Fig 4. Feeding of *H. variegata* males (A) and females (B) in the presence or absence of different compounds' odors at the concentration of 10 μg/μL.** Asterisks (*) denote statistically significant differences at $p<0.05$, while double asterisks (**) denote highly significant differences at $p<0.01$. and statistically significant differences, and ns denotes no significant difference.

differences were observed in males or with other compounds. In EAG tests, strength of the response to these four compounds increased with increasing concentration. Of these four compounds, the highest responses by the lady beetle were to octyl formate and ethyl benzoate. Finally, through the evaluation of predator feeding in the presence or absence of volatiles, we found that ethyl benzoate, octyl formate, methyl jasmonate, and methyl dihydrojasmonate promote greater feeding by *H. variegata*. The above tests showed that these volatiles not only attract *H. variegata* but also promote its feeding.

These four substances, which had an attractive effect on *H. variegata*, are all esters released by plants [39–43]. The volatile compounds released by plants themselves are relatively environmentally friendly. Among them, methyl jasmonate attracted adult *H. axyridis* at concentrations of 1000 μg/mL and 10,000 μg/mL [23]. This finding high-lighted the substantial potential of developing methyl jasmonate as a broad-spectrum attractant, capable of luring two predatory natural enemy species, thereby enhancing pest control efficacy across diverse pest populations. While many previous studies on plant volatiles have focused solely on insect olfactory responses, this research further investigated how volatiles attracting *H. variegata* influenced its predatory efficiency. However, our findings remained preliminary, and field application tests are required for validation. Moreover, practical field attractants were rarely single-component formulations but rather combinations of multiple substances. In subsequent studies, we plan to further explore multi-component combinatorial effects to develop more efficient and stable natural enemy attractants.

In chemical ecology research, the use of concentration gradients (1, 10, and 100 μg/μL) for insect behavioral response testing is a widely adopted standard methodology. Previous studies had demonstrated that insects exhibit significant concentration dependency in their responses to compounds, and multidose testing serves as a critical approach for assessing dose–response relationships [44]. Furthermore, insect behavioral responses to compounds were characterized by an "optimal response concentration range," beyond which the responses weaken or even disappear [45]. These findings provided clear literature support and theoretical justification for the experimental design employed (using 1, 10, and 100 μg/μL concentrations), confirming the scientific rationality and comparability of the approach. Through multi-concentration behavioral assays, we observed that the four compounds of greatest promise consistently exhibited attraction effects at 10 and 100 μg/μL, indicating their robust dose effects and high potential for field applications. We believe these results have significant reference value for the subsequent development and promotion of attractants targeting *H. variegata*.

In the electroantennography experiment, the antennae of *H. variegata* responded strongly to octyl formate and ethyl benzoate. Nevertheless, the responses to methyl jasmonate and methyl dihydrojasmonate were minimal. However, in behavioral tests *H. variegata* was attracted to all four compounds. This difference between EAG responses and the beetle's behavioral responses is common in studies of insect [46]. Methyl jasmonate and methyl dihydrojasmonate act as volatile signaling compounds, triggering defense responses in intact plants and amplifying their volatile organic compound (VOC) emissions [40]. This systemic induction may partially account for the observed discrepancy between EAG responses and behavioral responses. EAG may only reflect the initial response of the antennal olfactory receptors, and not fully represent the comprehensive behavioral response of insects. Behavioral responses involve more complex physiological and cognitive processes, and whole animal responses may be related to the integration of smell, sight, and other senses [47]. Most olfactory research has focused on short-range directed behaviors [48], while some chemical cues may work at long distances and trigger more complex behaviors. Up to the present, we have only tested the short distances effects of the four identified volatile compounds on *H. variegata*. We cannot rule out the possibility that they may also have long distances effects. Although these compounds had already demonstrated attractive activity at the behavioral level, we believe that EAG testing was of significant value in verifying their olfactory recognition capabilities, assessing concentration-sensing characteristics, and supporting the interpretation of behavioral results. It served as strong corroboration and deepened our understanding of the findings from Y-tube olfactometer experiments.

Complex interactions, such as synergism, antagonism, or neutralization, might exist between different compounds. Previous studies had shown that certain individual components might have weak attractiveness on their own but could significantly enhance overall attraction when combined with other components. For example, the two main components of thrips aggregation pheromones, (R)-lavandulyl acetate and neryl (S)-2-methylbutanoate, did not attract their predatory natural enemy Orius laevigatus when used individually. However, when mixed in specific proportions, they exhibited significant attraction to both adults and nymphs [49]. This clearly demonstrated that behavioral regulation activity often depends on the combined effects of multiple components rather than the absolute activity of a single component. The four plant volatiles in this study may also exhibit synergistic effects when combined with other compounds. However, this study did

not further explore this possibility, which is a direction worthy of further investigation in the future. Mixtures of these four substances could be tested to determine whether they have an attractive effect. But use of mixtures would require field testing to determine the most suitable combinations. In subsequent experiments, we will prioritize observing their performance under field conditions, as examining compound mixtures could reveal potential synergistic effects. This potential control strategy provides a new way of thinking about pest control.

Using molecular methods, the mechanisms behind the responses of *H. variegata* could be elucidated to provide understanding at a deeper level. The complex olfactory system of insects possesses odorant-binding proteins (OBPs), which are capable of detecting odors. Odorant molecules can contact the OBP and bind to olfactory receptors, thereby inducing sensory neurons to generate electrical signals that are transmitted to the brain and trigger corresponding olfactory behavioral responses [50]. HvarOBP5 is associated with the plant volatiles hexyl caproate and geranyl acetate, which can strongly bind to this protein [51]; HvarOBP2 can bind with plant volatiles such as β-ionone, dibutyl phthalate, nerolol, and oleic acid [52]. These studies of odorant binding proteins improve our understanding of the olfactory mechanism by which *H. variegata* locates host plants and prey habitats. The compounds found to be effective at-tracts in this study should in future studies have their olfactory roles confirmed using these molecular mechanisms to provide a theoretical basis for the study of targeted attractants in *H. variegata*. In farmland ecosystems, aphids, foliar-feeding mites, and other small-bodied pests often initially have spotty distributions in the crop, and the time needed for natural enemies to colonize farm fields often results in delays between their population growth and that of the pests. Due to such time lags between pests and their natural enemies, pests often build to damaging levels before natural enemies can achieve high enough numbers to exert control. Attractants may speed up the process of colonization of localized patches of pests by natural enemies, resulting in faster, better control.

## Supporting information

**S1 Data. ZIP file containing raw experimental data and metadata for behavioral and electrophysiological responses of *Hippodamia variegata* to plant volatiles. File contents: (1) README_Hippodamia_Volatiles.txt – Description of experimental design, assay methods, data structure, and variable definitions; (2) H_variegata_ response_to_plant_volatiles.xls: Includes three sheets—  • Sheet 1: Y-tube olfactometer assay results for initial screening;  • Sheet 2: Y-tube assays for concentration-dependent behavior;  • Sheet 3: Electroantennography (EAG) responses;  • Sheet 4: Cage feeding bioassay data.**
(ZIP)

## Author contributions

**Conceptualization:** Yanhui Lu.

**Data curation:** Yanhui Lu.

**Formal analysis:** Yi Wang, Ying Zhang, Tao Zhang, Zihan Li.

**Funding acquisition:** Yanhui Lu.

**Investigation:** Yi Wang, Ying Zhang, Zihan Li.

**Methodology:** Yanhui Lu.

**Project administration:** Yanhui Lu.

**Resources:** Yanhui Lu.

**Software:** Yanhui Lu.

**Supervision:** Tao Zhang, Tinghui Liu, Yanhui Lu.

**Validation:** Yanhui Lu.

**Visualization:** Yanhui Lu.

**Writing – original draft:** Yi Wang, Ying Zhang.

**Writing – review & editing:** Tao Zhang, Tinghui Liu, Yanhui Lu.

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
