## [Decision Letter · Decision Letter 0]

Dear Dr. Lu,

Thank you for submitting your manuscript to PLOS ONE. After careful consideration, we feel that it has merit but does not fully meet PLOS ONE’s publication criteria as it currently stands. Therefore, we invite you to submit a revised version of the manuscript that addresses the points raised during the review process.

**I agree with the comments pointed out by the reviewers. Please improve the text according to the reviewers' comments.**

We look forward to receiving your revised manuscript.

Kind regards,

Yonggen Lou

Academic Editor

PLOS ONE

 [This work was supported by the National Key R&D Program of China (2022YFD1400300), and China Agriculture Research System of MOF and MARA (CARS-15-19).].

3. In the online submission form, you indicated that [Data will be made available on request.].

Additional Editor Comments (if provided):

Reviewers' comments:

Reviewer's Responses to Questions

**Comments to the Author**

1. Is the manuscript technically sound, and do the data support the conclusions?

Reviewer #1: Yes

Reviewer #2: Yes

2. Has the statistical analysis been performed appropriately and rigorously?

Reviewer #1: Yes

Reviewer #2: Yes

3. Have the authors made all data underlying the findings in their manuscript fully available?

Reviewer #1: No

Reviewer #2: Yes

4. Is the manuscript presented in an intelligible fashion and written in standard English?

Reviewer #1: Yes

Reviewer #2: Yes

Reviewer #1: In this study four volatile compounds — methyl dihydrojasmonate, methyl jasmonate, octyl formate, and ethyl benzoate — were identified as significant attractants for the ladybeetle H. variegata. Behavioral and electroantennography (EAG) assays confirmed their attractant effects, showing dose-dependent responses (stronger signals at higher concentrations). Moreover, these volatiles enhanced ladybeetle’s feeding activity in enclosed environments, supporting their potential use in pest management by boosting natural predator behavior.

My overall comments are as follow:

1. The results are overall clearly presented.

2. A regression analysis quantifying the EAG response-concentration relationship for each compound would strengthen the statistical interpretation (e.g., linear vs. nonlinear trends).

3. The role of methyl jasmonate and methyl dihydrojasmonate as volatile plant defense signals (e.g., induced emission in undamaged plants) and how this might explain discrepancies between EAG and behavioral assays should be mentioned in discussion. Relevant citations should be incorporated.

4. Original data used for statistics were not provided.

Minor comments

Line 128 and 129: the dash in these words “de-scribed” and “con-centration” are not necessary.

Line 142: “Beetle larvae were tested by also connecting one end of the antenna to the reference electrode and the other end to the recording electrode, as described above.” Clarify that larval EAG data were not included (to avoid confusion).

Line 158: “A. gossypii” should be in italic.

Line 159-160: Specify filter paper height (e.g., "10 cm above the soil surface").

Line 265-267: methyl jasmonate and methyl dihydrojasmonate are volatile signals that can induce defense responses in undamaged plants and enhance their volatile emission. This induction might also be one reason for the difference between EAG responses and the behavioral responses. The discussion should be improved and citations about this point should be included.

Figure 3. Add regression lines with R²/p-values to visualize volatile concentration-EAG response relationships.

How about the effect of volatile compound mixture? Future Work: testing compound mixtures could reveal synergistic effects (useful for field applications).

Reviewer #2: This study evaluated behavior activity of 28 representative volatile compounds to Hippodamia variegata, and found four chemicals including ethyl benzoate were significantly attractive to H. variegata. Furthermore, behavioral and EAG responses of different concentrations of four chemicals to H. variegata were measured, which indicated that higher concentrations of 10 and 100 μg/μL had a significant attraction effect on H. variegata. This paper showed us an interesting story, but there are some mistake. Some comments are as follows

1. line 69� ‘Mentha’ need to be italic.

2. line 126, it is better to introduce why select 100, 10, and 1 μg/μL these three concentration

3. line 158, 'A. gossypii' need to be italic.

4. line 205-206, between 100 and μg/μL, one space is be lack. please check other place in text.

5. line 'By comparing their P-values, ... have better attraction effects on the male H. variegata at 10 μg/μL than at 100μg/μL.' The size of the number of P-values between two treatment to CK could to be used to deduce the difference of those two events?

**Do you want your identity to be public for this peer review?** For information about this choice, including consent withdrawal, please see our Privacy Policy

Reviewer #1: No

Reviewer #2: No

---

## [Author Response · Author response to Decision Letter 1]

30 May 2025

Dear Editors and Reviewers:

Thanks for your letter and constructive comments concerning our manuscript entitled “Behavioral and electrophysiological responses of Hippodamia variegata to plant volatiles” (ID: PONE-D-25-17465). We greatly appreciate your complimentary comments and suggestions. Based on your suggestions, we have made corrected modifications throughout the manuscript.

We hope the present version meet the Journal’s approval. The point-to-point responses were listed below.

Sincerely,

Yanhui Lu

Reply: We have made modifications according to PLOS ONE's style.

[This work was supported by the National Key R&D Program of China (2022YFD1400300), and China Agriculture Research System of MOF and MARA (CARS-15-19).].

Reply: Thank you for your guidance regarding the Funding Statement. As requested, I have included the amended funding declaration within the cover letter below. Please let me know if any additional information is needed.

3. In the online submission form, you indicated that [Data will be made available on request.].

Reply: We will provide access to the raw experimental data underlying this manuscript, with detailed datasets available in the supplementary files.

Reviewer #1:

1. The results are overall clearly presented.

Reply: Thank you for your positive feedback on the clarity of the results presentation. We are pleased to know that the organization and interpretation of the data met the journal's standards.

If there are any specific sections within the Results that require further elaboration or refinement to enhance readability, please do not hesitate to let us know. We are fully prepared to make additional revisions as needed.

2. A regression analysis quantifying the EAG response-concentration relationship for each compound would strengthen the statistical interpretation (e.g., linear vs. nonlinear trends).

Reply: Thanks. The relationship between substance concentration and the olfactory response of the lady beetle may not be linear. In our experiment, we tested six concentration gradients. Behavioral data revealed that the selectivity of H. variegata did not consistently increase with rising concentrations. Specifically, their olfactory response did not necessarily intensify at higher concentrations, and beyond a certain threshold, the response may stabilize. Therefore, we conclude that adding regression lines with R²/p-values to the graph would be inappropriate.

3. The role of methyl jasmonate and methyl dihydrojasmonate as volatile plant defense signals (e.g., induced emission in undamaged plants) and how this might explain discrepancies between EAG and behavioral assays should be mentioned in discussion. Relevant citations should be incorporated.

Reply: We sincerely appreciate the valuable comments. We have incorporated the pertinent discussions into the revised manuscript and supplemented corresponding references to support these additions. L268-271.

4. Original data used for statistics were not provided.

Reply: We will provide access to the raw experimental data underlying this manuscript, with detailed datasets available in the supplementary files.

Minor comments

Line 128 and 129: the dash in these words “de-scribed” and “con-centration” are not necessary.

Reply: Thank you for your valuable suggestions. We have made the corresponding revisions in the manuscript.

Line 142: “Beetle larvae were tested by also connecting one end of the antenna to the reference electrode and the other end to the recording electrode, as described above.” Clarify that larval EAG data were not included (to avoid confusion).

Reply: Thank you for your feedback. This study did not involve any larval experiments. We have revised the relevant content in the manuscript.

Line 158: “A. gossypii” should be in italic.

Reply: Thanks. A. gossypii has been formatted in italics.

Line 159-160: Specify filter paper height (e.g., "10 cm above the soil surface").

Reply: Thank you for your suggestion. We have revised the sentence to “Subsequently, 20 μL of the test compound (10 μg/μL) was dropwise applied onto filter paper positioned 10 cm above the soil surface of an individual plant.” L160-161.

Line 265-267: methyl jasmonate and methyl dihydrojasmonate are volatile signals that can induce defense responses in undamaged plants and enhance their volatile emission. This induction might also be one reason for the difference between EAG responses and the behavioral responses. The discussion should be improved and citations about this point should be included.

Reply: Thanks. We have added this content to the discussion section and included relevant references. The added content is as follows: Methyl jasmonate and methyl dihydrojasmonate act as volatile signaling compounds, triggering defense responses in intact plants and amplifying their volatile organic compound (VOC) emissions. This systemic induction may partially account for the observed discrepancy between EAG responses and behavioral responses. L268-271.

Figure 3. Add regression lines with R²/p-values to visualize volatile concentration-EAG response relationships.

Reply: The relationship between substance concentration and the olfactory response of the lady beetle may not be linear. In our experiment, we tested six concentration gradients. Behavioral data revealed that the selectivity of H. variegata did not consistently increase with rising concentrations. Specifically, their olfactory response did not necessarily intensify at higher concentrations, and beyond a certain threshold, the response may stabilize. Therefore, we conclude that adding regression lines with R²/p-values to the graph would be inappropriate.

How about the effect of volatile compound mixture? Future Work: testing compound mixtures could reveal synergistic effects (useful for field applications).

Reply: Thank you for your question. The effects of volatile mixtures are indeed a key focus of our future research. In subsequent experiments, we will prioritize observing their performance under field conditions, as examining compound mixtures could reveal potential synergistic effects. We have also expanded the discussion section to address this topic. L297-298.

Reviewer #2:

1. line 69, ‘Mentha’ need to be italic.

Reply: Thanks. We have italicized ‘Mentha’.

2. line 126, it is better to introduce why select 100, 10, and 1 μg/μL these three concentration

Reply: In chemical ecology research, using concentration gradients of 1, 10, and 100 μg/μL to conduct insect behavioral response tests is a widely adopted standard method. For example, Jiménez-Santiago et al. (2024) employed dose gradients of 1, 10, and 100 ng in their study on the behavioral responses of Triatoma pallidipennis to volatile substances, systematically evaluating the effects of different concentrations on insect attraction and repellence in a Y-tube olfactometer. We have added the corresponding references in the manuscript. L126.

3. line 158, 'A. gossypii' need to be italic.

Reply: Thanks. We have italicized 'A. gossypii'.

4. line 205-206, between 100 and μg/μL, one space is be lack. please check other place in text.

Reply: Thank you for highlighting the minor formatting issues. We have made the necessary corrections and reviewed the entire article for similar minor issues.

5. line 'By comparing their P-values, ... have better attraction effects on the male H. variegata at 10 μg/μL than at 100μg/μL.' The size of the number of P-values between two treatment to CK could to be used to deduce the difference of those two events?

Reply: The p-value is a statistical parameter used to determine the results of hypothesis testing. It represents the probability of observing the current result or more extreme outcomes under the assumption that the null hypothesis is true. A smaller p-value provides stronger justification to reject the null hypothesis and conclude that differences exist between the compared groups. In this experiment, the p-value partially reflects the level of ladybug selection rate; generally, a smaller p-value indicates higher selectivity in ladybug behavior. Therefore, the p-values between the two experimental treatments and the control group (CK) can be directly compared to evaluate differences.

---

## [Decision Letter · Decision Letter 1]

Behavioral and electrophysiological responses of Hippodamia variegata to plant volatiles

PONE-D-25-17465R1

Dear Dr. Lu,

We’re pleased to inform you that your manuscript has been judged scientifically suitable for publication and will be formally accepted for publication once it meets all outstanding technical requirements.

Kind regards,

Yonggen Lou

Academic Editor

PLOS ONE

Additional Editor Comments (optional):

Reviewers' comments:

Reviewer's Responses to Questions

**Comments to the Author**

Reviewer #1: (No Response)

2. Is the manuscript technically sound, and do the data support the conclusions?

Reviewer #1: (No Response)

3. Has the statistical analysis been performed appropriately and rigorously?

Reviewer #1: (No Response)

4. Have the authors made all data underlying the findings in their manuscript fully available?

Reviewer #1: (No Response)

5. Is the manuscript presented in an intelligible fashion and written in standard English?

Reviewer #1: (No Response)

Reviewer #1: Figure 3. Nonlinear Regression will be helpful in this case.

'By comparing their P-values, ... have better attraction effects on the male H.

variegata at 10 μg/μL than at 100μg/μL.' A statistical analysis of the attraction of males to methyl jasmonate and methyl dihydrojasmonate between concentrations of 10 μg/μL and 100 μg/μL is necessary to support this statement.

"a p-value, or statistical significance, does not measure the size of an effect or the importance of a result"

Wasserstein, Ronald L.; Lazar, Nicole A. (2016-04-02). "The ASA Statement on p -Values: Context, Process, and Purpose". The American Statistician. 70 (2): 129–133. doi:10.1080/00031305.2016.1154108. ISSN 0003-1305. S2CID 124084622.

**Do you want your identity to be public for this peer review?** For information about this choice, including consent withdrawal, please see our Privacy Policy

Reviewer #1: No

---

## [Editor Report · Acceptance letter]

PONE-D-25-17465R1

PLOS ONE

Dear Dr. Lu,

I'm pleased to inform you that your manuscript has been deemed suitable for publication in PLOS ONE. Congratulations! Your manuscript is now being handed over to our production team.

Kind regards,

on behalf of

Dr. Yonggen Lou

Academic Editor

PLOS ONE